# Association between Anti-Erythropoietin Receptor Antibodies and Cardiac Function in Patients on Hemodialysis: A Multicenter Cross-Sectional Study

**DOI:** 10.3390/biomedicines10092092

**Published:** 2022-08-26

**Authors:** Yasuhiro Mochida, Akinori Hara, Machiko Oka, Kyoko Maesato, Kunihiro Ishioka, Hidekazu Moriya, Megumi Oshima, Tadashi Toyama, Shinji Kitajima, Yasunori Iwata, Norihiko Sakai, Miho Shimizu, Yoshitaka Koshino, Takayasu Ohtake, Sumi Hidaka, Shuzo Kobayashi, Takashi Wada

**Affiliations:** 1Department of Nephrology and Laboratory Medicine, Faculty of Medicine, Institute of Medical, Pharmaceutical and Health Sciences, Kanazawa University, Kanazawa 920-8641, Japan; 2Kidney Disease and Transplant center, Shonan Kamakura General Hospital, Kamakura, Kanagawa 247-8533, Japan; 3Division of Nephrology, Kanazawa University Hospital, Kanazawa 920-8641, Japan; 4Department of Hygiene and Public Health, Faculty of Medicine, Institute of Medical, Pharmaceutical and Health Sciences, Kanazawa University, Kanazawa 920-8640, Japan; 5Department of Nephrology, Shonan Fujisawa Tokushukai Hospital, Kanagawa 251-0041, Japan; 6Department of Nephrology, Tokyo Nishi Tokushukai Hospital, Tokyo 196-0003, Japan; 7Department of Internal Medicine, Mizuho Hospital, Kahoku 929-0346, Japan

**Keywords:** erythropoietin (EPO), erythropoietin receptor (EPOR), anti-erythropoietin receptor antibodies, hemodialysis (HD), left ventricular hypertrophy, cardiac function

## Abstract

Cardiac dysfunction is an important prognostic predictor of cardiovascular mortality in patients on hemodialysis (HD). Erythropoietin (EPO) has been reported to improve cardiac function by binding to the EPO receptor (EPOR) on cardiomyocytes. This study investigated whether anti-EPOR antibodies were associated with left ventricular cardiac function in patients undergoing HD. This multicenter, cross-sectional observational study included 377 patients (median age, 70 years; 267 (70.8%) males) with chronic kidney disease (CKD) undergoing stable maintenance HD. Serum levels of anti-EPOR antibodies were measured, and echocardiography was used to assess the left ventricular mass index (LVMI) and left ventricular ejection fraction (LVEF). Anti-EPOR antibodies were found in 17 patients (4.5%). LVMI was greater (median of 135 g/m^2^ vs. 115 g/m^2^, *p* = 0.042), and the prevalence of LVEF < 50% was higher (35.3% vs. 15.6%, *p* = 0.032) in patients with anti-EPOR antibodies than in those without. Multivariable linear regression and logistic regression analysis (after adjusting for known risk factors of heart failure) revealed that anti-EPOR antibodies were independently associated with LVMI (coefficient 16.2%; 95% confidence interval (CI) 1.0–35.0%, *p* = 0.043) and LVEF <50% (odds ratio 3.20; 95% CI 1.05–9.73, *p* = 0.041). Thus, anti-EPOR antibody positivity was associated with left ventricular dysfunction in patients undergoing HD.

## 1. Introduction

Cardiovascular disease (CVD) is a poor prognostic factor in patients on hemodialysis (HD) [1,2]. Heart failure and left ventricular hypertrophy in patients on HD, which have prevalences of 43% [3] and about 80% [4,5], respectively, are major CVD complications and risk factors for cardiovascular mortality [2,6,7]. Factors associated with heart failure and left ventricular hypertrophy in patients on HD include older age, diabetes mellitus (DM), a history of coronary artery disease, hypertension, impaired left ventricular ejection fraction (LVEF), and anemia [8,9]. Therefore, early detection and treatment of these factors are essential for improving the CVD prognosis. 

In patients with chronic kidney disease (CKD), including those undergoing HD, erythropoietin (EPO)-producing cells are dysfunctional, and anemia is caused by a decrease in the production of endogenous EPO. EPO-stimulating agents (ESAs) have been used to treat anemia in these patients and have also been reported to improve cardiac function [10]. In particular, ESA administration reduced left ventricular myocardial mass [11,12,13] and increased LVEF [14] in addition to improving anemia. Furthermore, animal studies in mice have shown that when EPO receptors (EPORs) are functional in cardiomyocytes, myocardial mass and cardiac function are maintained, while when EPORs are missing or not functional in cardiomyocytes, left ventricular hypertrophy develops and LVEF decreases despite increasing levels of EPO [15]. Thus, the cardioprotective function of ESA is suggested to be caused, in part, by the direct action of EPO binding to EPOR in heart tissue along with improvement in anemia [15,16].

Recently, we detected autoantibodies to EPOR that can induce anemia by inhibiting EPO–EPOR interaction on erythroid progenitor cells [17]. Thereafter, we also showed that these antibodies were associated with poor kidney outcomes in patients with CKD, including lupus nephritis [18], DM [19,20], and anti-neutrophil cytoplasmic antibody (ANCA)-associated vasculitis [21]. However, the association of these antibodies with cardiac function in patients with CKD or on HD remains unknown.

Here, we hypothesized that anti-EPOR antibodies are involved in the pathogenesis of cardiac dysfunction in patients with CKD. To clarify this hypothesis, the present study aimed to investigate the relationship between anti-EPOR antibodies and cardiac function in patients on HD.

## 2. Materials and Methods

### 2.1. Study Patients

This multicenter cross-sectional observational study included 489 patients on maintenance HD recruited from February 2018 to August 2020 at six dialysis centers: Kanazawa University Hospital (Kanazawa, Japan), Mizuho Hospital (Kahoku, Japan), Moriyama Koshino Clinic (Kanazawa, Japan), Shonan Kamakura General Hospital (Kanagawa, Japan), Shonan Fujisawa Tokushukai Hospital (Kanagawa, Japan), and Tokyo Nishi Tokushukai Hospital (Tokyo, Japan). After excluding patients who did not provide informed consent (85 patients), we enrolled the remaining 404 patients on HD (Figure 1). We collected data on comorbidities and clinical characteristics from electronic medical records and evaluated cardiac function using echocardiography. We excluded patients who did not undergo echocardiography (24 patients), as well as those who had a hematological disorder or a malignant tumor. Finally, 377 patients were enrolled as study participants.

### 2.2. Evaluation of Clinical Characteristics

We obtained information about baseline patient characteristics and medication use (renin–angiotensin system (RAS) inhibitors) from electronic medical records. Age, sex, dialysis duration, and comorbidities (hypertension (HT), dyslipidemia, DM, ischemic heart disease (IHD), stroke, and peripheral artery disease (PAD)) at baseline were collected. Body mass index (BMI) was calculated by body weight (kg) divided by height squared (m^2^). Before dialysis, both systolic and diastolic blood pressures were measured, and blood samples were collected. The following laboratory values were also collected: white blood cell count (WBC), hemoglobin (Hb), albumin, triglyceride (TG), low-density lipoprotein cholesterol (LDL-cho), high-density lipoprotein cholesterol (HDL-cho), C-reactive protein (CRP), and β2-microglobulin levels. The erythropoietin resistance index (ERI) was calculated using the following equation [22]: ERI = EPO dose/week/dry weight (kg) ×Hb (g/dL). The EPO dose was calculated by converting the weekly epoetin-α dose. If darbepoetin-α or continuous erythropoietin receptor activators were used as ESAs, these EPO doses were multiplied by 200 to convert them to doses of epoetin-α in units [22].

### 2.3. Measurement of Anti-EPOR Antibodies

Serum levels of anti-EPOR antibodies were measured by enzyme-linked immunosorbent assay, as previously described [17]. Briefly, polyvinyl 96-well microplates (Nunc International, Tokyo, Japan) were coated with recombinant human EPOR protein (R&D Systems, Minneapolis, MN, USA) diluted in 0.2 M sodium bicarbonate buffer and incubated at 4 °C for 24 h. The remaining free binding sites were blocked with 1% bovine serum albumin at 4 °C. After the plates were washed with Tween 20-Tris buffered saline, serum samples were added in duplicate at 1:1000 dilution to 1% bovine serum albumin in phosphate-buffered saline for 20 h at 4 °C. The plates were washed with the same buffer and incubated with goat anti-human IgG conjugated with horseradish peroxidase (Millipore, Temecula, CA, USA) at a dilution of 1:5000 for 1.5 h at room temperature. The substrate, tetramethylbenzidine (KPL, Gaithersburg, MD, USA), was added, and the reaction was stopped by adding 2N sulfuric acid. The optical density (OD) at 450 nm was determined by an automated plate reader. Anti-EPOR antibodies were considered positive when the OD450 ratio between patient serum and normal control serum was greater than 1.5 [23].

### 2.4. Echocardiography

Standard echocardiography recordings were performed on midweek non-dialysis days by two expert echocardiographers and/or doctors at the participating institutions. We assessed LVMI and LVEF as cardiac function markers. LVMI was calculated by dividing the left ventricular mass by the body surface area. The left ventricular mass was calculated from the left ventricular end-diastolic diameter (LVEDD), end-systolic diameter (LVESD) as well as the posterior wall diameter (PWd), and interventricular septum diameter (IVSd) from the parasternal long axis position using the formula by Devereux [24] with the recommendations of the American Society of Echocardiography [25]: left ventricular mass = 0.8 × 1.04 × [(LVEDD + IVSd + PWd)^3^-LVEDD^3^] + 0.6. LVEF was calculated from the ratio of the left ventricular stroke volume (SV) to the left ventricular end-diastolic volume (LVEDV), and SV was obtained by subtracting the left ventricular end-systolic volume (LVESV) from LVEDV [26]; LVEF = (LVEDV − LVESV) × 100 (%)/LVEDV.

### 2.5. Statistical Analysis

The patients were divided into two groups: those who had anti-EPOR antibodies and those who did not, and those with or without LVEF < 50%. Nominal variables were expressed as percentages. After using the Shapiro–Wilk test for normality distribution, data for continuous variables were expressed as means ± standard deviations (SD) for normally distributed data or medians (interquartile ranges: IQR) for non-normally distributed data. To compare the two groups, we used Fisher’s exact test for nominal variables, Student’s *t*-test for continuous variables with a normal distribution, and Mann–Whitney’s U-test for a non-normal distribution. Multivariable linear regression for log-LVMI and logistic regression analyses for LVEF <50% were also performed to determine whether the presence of anti-EPOR antibodies was independently associated with each left ventricular function parameter, adjusting for age, sex, IHD, DM, RAS inhibitor use, BMI, systolic blood pressure, Hb, dyslipidemia Alb, CRP, and dialysis duration as known risk factors of cardiac dysfunction in this population [9,27,28]. The levels of LVMI as the dependent variable in multivariable linear regression were natural log-transformed as log-LVMI to provide normality in the regression residuals. Beta coefficients in the linear regression analysis for log-LVMI were back-transformed to indicate the percent increase in LVMI with one unit increase in the corresponding covariate. Stata ver.15 (Stata Corp LLC, College Station, TX, USA) was used for all statistical analyses. A *p*-value less than 0.05 was considered statistically significant.

## 3. Results

### 3.1. Patient Characteristics

Table 1 shows the patient characteristics of the 377 enrolled patients. The median age was 70 years, the proportion of males was 70.8%, 51.7% had DM, 26.3% had IHD, and the median dialysis duration time was 60 months (IQR 27–124 months). Anti-EPOR antibodies were detected in 17 patients (4.5%). Of these, antibodies were observed in 10 of 195 patients (5.1%) with DM and 7 of 182 patients (3.8%) without DM. The causes of renal failure in patients with anti-EPOR antibodies were diabetic kidney disease in eight cases (47%), nephrosclerosis in two (11.8%), chronic glomerulonephritis in three (17.6%), polycystic kidney disease in one (5.9%), gouty kidney in one (5.9%), Alport syndrome in one (5.9%), and unknown in one (5.9%).

### 3.2. Clinical Characteristics of Patients with and without Anti-EPOR Antibodies

The demographic and clinical findings of patients with and without anti-EPOR antibodies are shown in Table 1. Patients with anti-EPOR antibodies had significantly lower levels of Hb (median (IQR) of 10.5 (9.6–11.0) g/dL vs. 11.1 (10.2–12.0) g/dL, *p* = 0.009), higher levels of CRP (0.33 (0.12–1.83) mg/dL vs. 0.15 (0.047–0.53) mg/dL, *p* = 0.028), higher levels of ERI (13.00 (5.38–21.65) vs. 5.75 (2.30–11.57), *p* = 0.026), and higher doses of ESA (7500 (2500–12,000) units/weekly vs. 4000 (1250–7500) units/weekly, *p* = 0.049) than those without antibodies. In the assessment of cardiac function, the anti-EPOR antibody-positive group had significantly higher levels of LVMI (135 (116–170) g/m^2^ vs. 115 (95–141) g/m^2^, *p* = 0.044, Figure 2A) and a higher proportion of patients with LVEF <50% (35.3% vs. 15.6%, *p* = 0.032) compared to those in the antibody-negative group (Figure 2B).

### 3.3. Clinical Characteristics of Patients with and without LVEF < 50%

The demographic and clinical findings of patients with and without LVEF < 50% are shown in Table 2. LVEF < 50% was observed in 62 patients (16.4%). LVEF < 50% was associated with a significantly higher proportion of anti-EPOR antibody positivity (9.7% vs. 3.5%, *p* = 0.044), male sex (85.5% vs. 67.9%, *p* = 0.006), DM (64.5% vs. 49.2%, *p* = 0.036), and IHD (51.6% vs. 21.3%, *p* < 0.001) compared to LVEF ≥ 50%. In addition, higher levels of CRP (0.22 (0.084–0.63) mg/dL vs. 0.14 (0.04–0.54) mg/dL, *p* = 0.016) and lower levels of HDL cholesterol (41 (34–46) mg/dL vs. 45 (35–60) mg/dL, *p* = 0.016) were observed in patients with LVEF < 50% compared with those without. Patients with LVEF < 50% had higher LVMI (141 (125–169) g/m^2^ vs. 112 (92–1235) g/m^2^, *p* < 0.001) than patients with LVEF ≥ 50%.

### 3.4. Association between the Presence of Anti-EPOR Antibodies and LVMI

In addition to the univariable analysis, multivariate linear regression analysis for log-LVMI showed that anti-EPOR antibodies were significantly associated with LVMI (back-transformed beta coefficient 16.7%; 95% confidence interval (CI) 0.7–35.2%, *p* = 0.043) as well as male sex and CRP levels (Table 3).

### 3.5. Association between the Presence of Anti-EPOR Antibodies and Reduced LVEF

We further performed multivariable logistic regression analysis with the same covariates as those used in model 1 of the multivariable linear regression analysis for LVMI. The presence of anti-EPOR antibodies, as well as male sex and IHD, was independently associated with reduced LVEF (odds ratio 3.20; 95% CI 1.05–9.73, *p* = 0.041) (Table 4).

## 4. Discussion

This multicenter cross-sectional observational study investigated the association between anti-EPOR antibodies and cardiac function in patients with CKD on maintenance HD. We found that positivity for anti-EPOR antibodies was independently associated with LVMI and reduced LVEF even after adjustment for known risk factors, including IHD, HT, and DM.

The proportion of patients positive for anti-EPOR antibodies in this study was 4.5%, which is smaller than found in patients with CKD in previous studies that included some causes of CKD with lupus nephritis [18], DM [19,20], ANCA vasculitis [21], and maintenance HD [23]. In particular, of these studies, Hara et al. showed that the proportion of anti-EPOR antibodies in patients on maintenance HD with ESA administration was 10.2% [23]. The difference in the proportion of anti-EPOR antibodies may be explained, in part, by the difference in the indication for the reason for hemodialysis and the fact that patients were enrolled regardless of using ESAs and their levels of hemoglobin.

Patients with anti-EPOR antibodies were treated with the administration of more doses of ESAs, while they had a lower concentration of hemoglobin than those without anti-EPOR antibodies in this study. These characteristics of ESA hyporesponsiveness in positive patients with anti-EPOR antibodies are consistent with the results of a previous study by Hara et al. [23]. Immunoglobulin G fractions from patients with anti-EPOR antibodies have been reported to inhibit the proliferation of a cultured erythroid cell line by blocking the EPO-EPOR binding pathway in in vitro experiments [17]. Whether anti-EPOR antibodies are associated with the development or progression of ESA hyporesponsive anemia in patients on dialysis needs to be investigated by further longitudinal studies.

The present study shows that male sex, IHD, and CRP levels were associated with LVMI or EF < 50%. The clinical factors of male sex and IHD as covariates are consistent with factors of left ventricular dysfunction in previous studies [8,9,29,30], as well as DM, hypertension, and hypoalbuminemia. On the other hand, our result showing that CRP was negatively associated with LVMI differs from previous reports. Besides the classical risk factors related to left ventricular dysfunction, as mentioned above, various risk factors, such as volume overload, CKD–Mineral and Bone Disorder (CKD-MBD), and Malnutrition–Inflammation–Atherosclerosis (MIA) syndrome, are known to be involved in the pathogenesis of cardiac dysfunction in patients on HD [31]. Collectively, the relationship between LVMI and inflammatory conditions in patients on HD needs to be further investigated using other inflammatory markers as well.

To the best of our knowledge, this is the first study showing that anti-EPOR antibodies are independently associated with LVMI and decreased LVEF. The following mechanisms might explain this association: First, anti-EPOR antibodies may inhibit the action of EPO binding to EPOR in cardiac tissues as well as bone marrow erythroid precursors. EPORs have been reported to be expressed not only on immature erythrocyte membranes but also on cardiomyocytes [15,32,33], and ESA administration has been shown to protect cardiac function through EPORs on cardiomyocytes after a cardiac injury [33]. In experimental mouse models in which EPOR is lacking in non-hematopoietic cells, left ventricular myocardial stress was found to lead to significantly greater left ventricular mass and less cardiac contractility than those in control mice, concomitant with decreased signal transducer and activator of transcription 3 (STAT3) signaling required for cardioprotection [15]. Second, the effect of the anti-inflammatory action of erythropoietin [34,35] might be blocked by anti-EPOR antibodies. The inflammatory milieu has been reported to cause the downregulation of EPOR [36], hyporesponsiveness of ESA [37], and increase in fibroblast growth factor 23 [36], which induced myocardial hypertrophy [38,39] and diastolic dysfunction [40]. Recently, patients on HD with high levels of ERI have been reported to be associated with cardiovascular events and mortality [37,41,42] in addition to chronic inflammation [37]. In our study, the anti-EPOR antibody-positive group had significantly higher CRP levels and higher ERI compared to the antibody-negative group. These studies and our data suggest that anti-EPOR antibodies inhibit the binding of EPO to EPOR and then suppress the signaling for cardioprotection, resulting in left ventricular hypertrophy and a reduction in cardiac systolic contractility in patients on HD.

There are several limitations to this study. First, echocardiography was not performed by the same physician or technologist or using the same device. Second, this study is an observational study reporting only data on associations from a small number of anti-EPO antibody-positive patients, so causative conclusions could not be drawn. In addition, owing to the nature of cross-sectional studies, the causal relationship between anti-EPOR antibodies and an increase in LVMI or reduced LVEF is unknown. Finally, the frequency of anti-EPOR antibody-positive patients and those with LVEF < 50% was small. As a result, known or possible covariates associated with cardiac dysfunction might not be fully included in the multivariable analysis. In addition, some of the statistical differences are a bit marginal though significant, and this may be attributed to the small number of patients in the specific group only.

## 5. Conclusions

Anti-EPOR antibody positivity was associated with left ventricular hypertrophy and systolic dysfunction in patients with CKD on HD. Whether the presence of anti-EPOR antibodies can cause cardiac dysfunction and subsequent cardiovascular mortality in similar populations needs to be further examined.

## Figures and Tables

**Figure 1 biomedicines-10-02092-f001:**
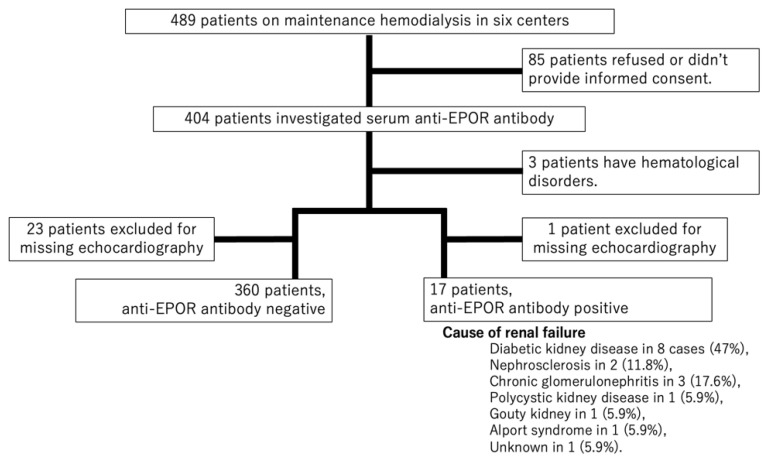
Patient selection flow chart.

**Figure 2 biomedicines-10-02092-f002:**
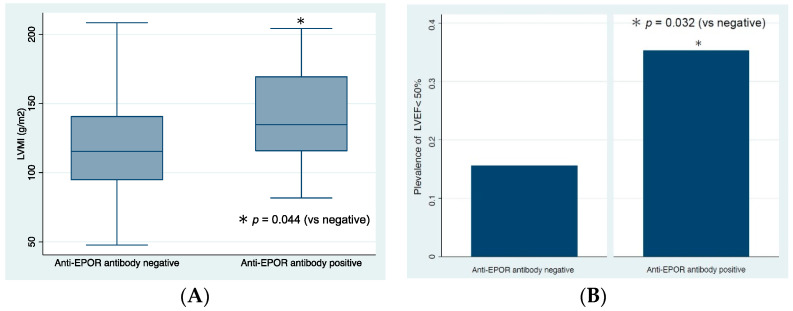
Cardiac function of patients with and without anti-EPOR antibodies. (**A**) Levels of left ventricular mass index (LVMI) and (**B**) prevalence of left ventricular ejection fraction (LVEF). EPOR, erythropoietin receptor.

**Table 1 biomedicines-10-02092-t001:** Baseline characteristics of patients with and without anti-EPO receptor antibodies.

		Anti-EPOR Antibodies		
	Total (*n* = 377)	Negative (*n* = 360)	Positive (*n* = 17)	*p*-Value
Age, y	70 (60–77)	70 (60–77)	73 (67–79)	0.22
Male, %	267 (70.8)	255 (70.8)	12 (70.6)	1.0
Dialysis duration, months	60 (27–124)	60 (27–125)	48 (27–79)	0.21
Comorbidity				
DM, %	195 (51.7)	185 (51.4)	10 (58.8)	0.62
HT, %	357 (94.7)	340 (94.4)	17 (100.0)	1.00
Dyslipidemia, %	351 (93.1)	335 (93.1)	16 (94.1)	1.00
Vascular disease, %	186 (49.3)	179 (49.7)	7 (41.2)	0.62
IHD, %	99 (26.3)	93 (25.8)	6 (35.3)	0.4
stroke, %	54 (14.3)	54 (15.0)	0 (0.0)	0.15
PAD, %	82 (21.8)	79 (21.9)	3 (17.6)	1.0
BMI, kg/m^2^	21.3 (19.0–23.8)	21.3 (19.1–23.9)	21.5 (17.8–22.8)	0.33
RAS inhibitor, %	217 (57.6)	209 (58.1)	8 (47.1)	0.45
SBP, mmHg	144 ± 24	144 ± 24	142 ± 24	0.69
DBP, mmHg	75 ± 13	75 ± 13	76 ± 15	0.70
Pulse pressure, mmHg	69 ± 18	69 ±18	65 ± 13	0.42
WBC, /µL	5400 (4300–6900)	5400 (4325–6900)	5700 (4300–6700)	0.95
RBC, ×10^4^/µL	365 ± 53	365 ± 53	345 ± 542	0.12
MCV, fL	94.2 ± 7.1	94.2 ± 7.2	93.3 ± 5.1	0.75
Hb, g/dL	11.0 (10.2–11.9)	11.1 (10.2–12.0)	10.5 (9.6–11.0)	0.009
Albumin, g/dL	3.5 (3.3–3.7)	3.5 (3.3–3.7)	3.3 (2.9–3.7)	0.053
Triglyceride, mg/dL	89 (64–123)	90 (63–125)	82 (72–116)	0.85
LDL-Cho, mg/dL	78 (63–97)	78 (63–98)	72 (57–80)	0.12
HDL-Cho, mg/dL	43 (35–58)	43 (35–58)	40 (34–49)	0.42
CRP, mg/dL	0.16 (0.049–0.54)	0.15 (0.047–0.53)	0.33 (0.12–1.83)	0.028
β2-Microglobulin, µg/L	27.5 ± 6.1	27.5 ± 6.1	25.7 ± 6.0	0.45
ERI	6.10 (2.32–12.25)	5.75 (2.30–11.57)	13.00 (5.38–21.65)	0.026
EPO dose	4000 (1350–7500)	4000 (1250–7500)	7500 (2500–12,000)	0.049

DM, diabetes mellitus; HT, hypertension; IHD, ischemic heart disease; PAD, peripheral artery disease; BMI, body mass index; DW, dry weight; RAS, renin–angiotensin system; SBP, systolic blood pressure; DBP, diastolic blood pressure; WBC, white blood cells; Hb, hemoglobin; -Cho, cholesterol; HDL, high-density lipoprotein; LDL, low-density lipoprotein; CRP, C-reactive protein; ERI, erythropoietin resistance index; EPO dose, erythropoietin dose calculated by converting weekly epoetin-α dose.

**Table 2 biomedicines-10-02092-t002:** Baseline characteristics of patients with and without LVEF < 50%.

	LVEF ≥ 50%	LVEF < 50%	*p*-Value
	*n* = 315	*n* = 62	
Anti-EPOR antibody, %	11 (3.5)	6 (9.7)	0.044
Age, y	70 (60–78)	69 (62–77)	0.49
Male, %	214 (67.9)	53 (85.5)	0.006
Dialysis duration, months	63 (28–128)	52 (22–111)	0.078
Comorbidity			
DM, %	155 (49.2)	40 (64.5)	0.036
HT, %	296 (94.3)	60 (96.8)	0.55
Dyslipidemia, %	295 (93.7)	56 (90.3)	0.41
Vascular disease, %			
IHD, %	67 (21.3%)	32 (51.6%)	<0.001
stroke, %	46 (14.6%)	8 (12.9%)	0.84
PAD, %	68 (21.6%)	14 (22.6%)	0.87
BMI, kg/m^2^	21.4 (19.0–23.7)	21.3 (19.0–24.7)	0.6
RAS inhibitor, %	151 (47.9)	23 (37.1)	0.13
SBP, mmHg	144 ± 24	144 ± 27	0.97
DBP, mmHg	75 ± 13	77 ± 15	0.26
Pulse pressure, mmHg	69 ± 18	67 ± 18	0.42
WBC, /µL	5500 (4400–6800)	4950 (3730–7100)	0.36
RBC, ×10^4^/µL	361 ± 48	383 ± 71	0.002
MCV, fL	94.3 ± 7.0	93.3 ± 7.8	0.38
Hb, g/dL	11.0 (10.2–11.8)	11.3 (10.5–12.1)	0.18
Albumin, g/dL	3.5 (3.3–3.8)	3.5 (3.1–3.7)	0.057
Triglyceride, mg/dL	86 (63–123)	101 (72–125)	0.14
LDL-Cho, mg/dL	78 (63–98)	76 (63–92)	0.34
HDL-Cho, mg/dL	45 (35–60)	41 (34–46)	0.016
CRP, mg/dL	0.14 (0.04–0.52)	0.215 (0.084–0.63)	0.016
β2-microglobulin, µg/L	27.9 (23.8–31.0)	26.9 (22.6–32.1)	0.64
ERI	6.12 (2.45–12.15)	5.91 (1.49–12.61)	0.61
EPO dose	4000 (2000–7500)	4000 (1000–8000)	0.82
LVMI, g/m^2^	112 (92–1235)	141 (125–169)	<0.001

LVEF, left ventricular ejection fraction; DM, diabetes mellitus; HT, hypertension; IHD, ischemic heart disease; PAD, peripheral artery disease; BMI, body mass index; DW, dry weight; RAS, renin–angiotensin system; SBP, systolic blood pressure; DBP, diastolic blood pressure; WBC, white blood cells; Hb, hemoglobin; -Cho, cholesterol; HDL, high-density lipoprotein; LDL, low-density lipoprotein; CRP, C-reactive protein; ERI, erythropoietin resistance index; EPO dose, erythropoietin dose calculated by converting weekly epoetin-α dose.

**Table 3 biomedicines-10-02092-t003:** Univariable and multivariable linear regression analysis for logarithmically transformed LVMI.

	Univariable		Multivariable Model 1		Multivariable Model 2	
	Coefficient (95%CI)	*p*	Coefficient (95%CI)	*p*	Coefficient (95%CI)	*p*
Presence of anti-EPOR antibody	16.7% (0.6–35.4%)	0.041	15.9% (0.03–35.0%)	0.049	16.7% (0.7–35.2%)	0.043
Age, y	0.2% (–0.4–0.1%)	0.169	−0.2% (−0.5–0.1%)	0.129	−0.2% (−0.5–0.1%)	0.126
Male	9.4% (2.0–16.2%)	0.007	8.3% (12.0–15.0%)	0.021	10.2% (2.0–17.4%)	0.012
Dialysis duration, months	1.0% (1.0–1.0%)	0.919	-	-	1.0% (1.0–1.0%)	0.334
DM	7.3% (1.0–13.9%)	0.018	6.0% (−1.0–12.7%)	0.074	6.2% (−1.0–12.7%)	0.082
IHD	6.2% (−1.0–12.7%)	0.114	4.9% (−2.0–12.7%)	0.178	5.1% (−2.0–12.7%)	0.197
BMI, kg/m^2^	0.1% (−1.0–0. 1%)	0.900	−0.5% (−13–0.3%)	0.255	−0.3% (−10.5–0.6%)	0.531
RAS inhibitor use	6.2% (0–12.7%)	0.055	-	-	4.1% (−2.0–10.5%)	0.187
SBP, mmHg	0.1% (0.0–0.2%)	0.032	0.1% (0.0–0.2%)	0.069	0.1% (−0.1–0.2%)	0.094
Hb, g/dL	0.2% (−0.2–2.5%)	0.85	-	-	1.0% (−2.0–3.0%)	0.607
Alb, d/dL	−0.3% (−10.2–5.1%)	0.487	-	-	−7.3% (–17.4–1.0%)	0.090
Dyslipidemia	−7.3% (−20.9–5.1%)	0.251	-	-	−4.1% (–17.4–8.3%)	0.475
CRP, mg/dL	−1.5% (−3.7–0.7%)	0.186	-	-	−3.0% (–5.1–−0.2%)	0.035

Beta coefficients were back-transformed to indicate the percent increase in LVMI with one unit increase in the corresponding covariate. The presence of anti-EPOR antibody induced a 16.7% increase in LVMI. LVMI, left ventricular mass index; CI, confidence interval; EPOR, erythropoietin receptor; DM, diabetes mellitus; IHD, ischemic heart disease; BMI, body mass index; RAS, renin–aldosterone system; SBP, systolic blood pressure; Hb, hemoglobin; Alb, albumin; HDL, high-density lipoprotein; CRP, C-reactive protein.

**Table 4 biomedicines-10-02092-t004:** Univariable and multivariable logistic regression analysis for reduced ejection fraction (LVEF < 50%).

	Univariable			Multivariable		
Covariates	OR	95% CI	*p*-Value	OR	95% CI	*p*-Value
Presence of anti-EPOR antibody	2.96	1.05–8.33	0.040	3.20	1.05–9.73	0.041
Age, y	0.99	0.97–1.01	0.433	0.98	0.95–1.00	0.123
Male	2.78	1.32–5.86	0.007	2.32	1.06–5.05	0.035
BMI, kg/m^2^	1.04	0.97–1.12	0.222	1.01	0.94–1.09	0.340
IHD	3.95	2.24–6.96	<0.001	3.73	2.03–6.84	<0.001
DM	1.88	1.07–3.30	0.029	1.46	0.79–2.70	0.225
SBP, mmHg	1.00	0.99–1.01	0.972	1.00	0.99–1.01	0.970

LVEF, left ventricular ejection fraction; OR, odds ratio; CI, confidence interval; EPOR, erythropoietin receptor; Hb, hemoglobin; IHD, ischemic heart disease; DM, diabetes mellitus; CRP, C-reactive protein; SBP, systolic blood.

## Data Availability

Data used in this study are available on request to the corresponding authors.

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
