# Peer review of "Association between Anti-Erythropoietin Receptor Antibodies and Cardiac Function in Patients on Hemodialysis: A Multicenter Cross-Sectional Study"

_biomedicines, 2022, doi:10.3390/biomedicines10092092_

Round 1

Reviewer 1 Report

    Authors examined the anti-EPOR in 377 patients and correlated the data with their cardiac function. The work was well performed and clearly presented. Comments are as the followings.

1. The anti-EPOR was only present in 4.5% of patients in this study. Readers would like to know if there is difference among other ethnic groups. Please provide such information if it's already known. Further, why the antibody is only present in such a small percentage of patients, since anemia is almost universal in HD patients.

2. Authors did not provide enough scientific information concerning the biological effect of this antibody, and also the mechanism. 

3. Authors noted that anti-EPOR was correlated with cardiac function. This study recruited patients till August, 2020 -- two years from now. Have authors followed their cardiovascular outcome? It will certainly be more persuasive if the positive anti-EPOR is associated with CV outome.

4. As authors mentkioned in your text, the (+) patients had received higher dose of EPO but showed lower level of Hb. However, the statistical analysis is insignificant, which is a pity. Is there any way to improve?

Author Response

Comments and Suggestions for Authors
    Authors examined the anti-EPOR in 377 patients and correlated the data with their cardiac function. The work was well performed and clearly presented. Comments are as the followings.
1. The anti-EPOR was only present in 4.5% of patients in this study. Readers would like to know if there is difference among other ethnic groups. Please provide such information if it's already known. 

→Thank you for pointing out the issue. As you pointed out, the percentage of Japanese subjects positive for EPO receptor antibody in this study is small, at 4.5%. The positive rate of the antibody in other racial groups is unknown at this time. However,  a recent study has reported the effect of this antibody on renal outcomes in participants of the ADVANCE and ADVANCE-ON studies, both of which are global clinical trials for type 2 diabetes. (Oshima M, et al. KI reports 2021).

Further, why the antibody is only present in such a small percentage of patients, since anemia is almost universal in HD patients.

→Thank you for pointing out the issue. As you mentioned, hemodialysis patients have anemia due to many kinds of causes, including deficiency of iron, vitamins, zinc, and copper, hyperparathyroidism, and uremia for lowefficiency of dialysis except for deficiency of endogenous EPO.( Yamamoto H, et al. Renal Replace Ther. 2017;3:36., KDIGO. Chapter 3 Kidney Int Suppl. 2012;2:299–310.)
In addition to those classical factors, the antibody can be a cause of anemia in CKD patients. At present, the mechanism by which these antibodies are produced is unknown (Hara A, et al. Clin Exp Nephrol 2020 Vol. 24 Issue 1 Pages 88-95), so it is difficult to address the detail of this small percentage seen in hemodialysis patients. This is an issue for further investigation.

2. Authors did not provide enough scientific information concerning the biological effect of this antibody, and also the mechanism. 

→Thank you for pointing this issue. We agree that information concerning the biological effect of this antibody, and also the mechanism have not provided enough. Based on the comment, we have added the following sentences in red color sentences in the Discussion section;
(page 8, line 237-239) Immunoglobulin G fractions from patients with anti-EPOR antibodies have been reported to inhibit the proliferation of a cultured erythroid cell line by blocking the EPO-EPOR binding pathway in in-vitro experiments.
(page 8, line 255-257) The following mechanisms might explain this association: First, anti-EPOR antibodies may inhibit the action of EPO to bind EPOR in cardiac tissues as well as bone marrow erythroid precursors.

3. Authors noted that anti-EPOR was correlated with cardiac function. This study recruited patients till August, 2020 -- two years from now. Have authors followed their cardiovascular outcome? It will certainly be more persuasive if the positive anti-EPOR is associated with CV outcome.

→Thank you very much for pointing out this important issue. Now, we are following up patients enrolled in the present study to evaluate the cardiovascular outcome in near future. 

4. As authors mentioned in your text, the (+) patients had received higher dose of EPO but showed lower level of Hb. However, the statistical analysis is insignificant, which is a pity. Is there any way to improve?

→Thank you for pointing this issue. We also think that these findings may reflect erythropoiesis-stimulating agents (ESA) hyporesponsiveness in patients with anti-EPOR antibodies; compared to the antibody-negative group, the antibody-positive group did not achieve the target hemoglobin concentration despite the use of higher doses of ESAs. These findings reflected the higher ERIs in the antibody-positive group. The present results are consistent with a previous report (Hara A, et al. Clin Exp Nephrol 2020 Vol.24 Issue 1 Pages88-95) showing an association between the antibodies and ESA hyporesponsiveness. Therefore, we appreciate if you let us to present these findings as original ones.

Reviewer 2 Report

Dear Authors, I found your study complete and well-written. However, I think that in the discussion it should be further emphasized that this is an observational study reporting only data on (a weak) association on a small number of anti-EPO Ab positive patients, so causative conclusions might not be drawn. Moreover, to better evaluate the impact of your findings, I suggest changing Figure 1a, substituting colons with individual patient data.

Author Response

Reviewer2
Comments and Suggestions for Authors
Dear Authors, I found your study complete and well-written. However, I think that in the discussion it should be further emphasized that "this is an observational study reporting only data on (a weak) association on a small number of anti-EPO Ab positive patients, so causative conclusions might not be drawn". 

→Thank you very much for pointing out this important issue. We added the sentence "this is an observational study reporting only data on (a weak) association on a small number of anti-EPO Ab positive patients, so causative conclusions might not be drawn " as a limitation in the Discussion section (P8L276-278).

Moreover, to better evaluate the impact of your findings, I suggest changing Figure 1a, substituting colons with individual patient data.

→We added the underlying disease of individual patients' data in Figure1.

Reviewer 3 Report

In the manuscript entitled "Association between Anti-erythropoietin Receptor Antibodies and Cardiac Function in Patients on Hemodialysis: A Multicenter Cross-Sectional Study" Yasuhiro Mochida et al. aim to assess the relation between Anti-EPOR antibodies and clinical manifestations of cardiac dysregulation in the context of HD patients. For this purpose, the authors have organized a detailed study and support their hypothesis with the use of classical yet elegant statistical analyses. In addition, and to their honor, the authors discuss the limitations of this study and their differences with other reported studies. Their findings can be useful in terms of clinical manifestations and monitoring of them.

This Reviewer only has some minor comments, as follow:

1.       Please check throughout the manuscript for spelling mistakes and for sporadically missing abbreviations. In addition, it would be easier for the reader to follow if the use of abbreviations (in specific sentences) could be somehow decreased.

2.       Do the authors have more RBC measurements (e.g. RBC count etc.) apart from cellular Hb? It would be very informative in the anemia context of these subjects.

3.       My main concern is the small number of antibody-positive subjects that can jeopardize the final statistical outcome. In fact, some of the differences are a bit marginal (e.g. p=0.044) nevertheless significant. Can this be attributed o the small number of the specific group only or do the authors have another explanation (clinical or statistical)? Please further comment about this in the discussion section.

Author Response

Reviewer3
Comments and Suggestions for Authors
In the manuscript entitled "Association between Anti-erythropoietin Receptor Antibodies and Cardiac Function in Patients on Hemodialysis: A Multicenter Cross-Sectional Study" Yasuhiro Mochida et al. aim to assess the relation between Anti-EPOR antibodies and clinical manifestations of cardiac dysregulation in the context of HD patients. For this purpose, the authors have organized a detailed study and support their hypothesis with the use of classical yet elegant statistical analyses. In addition, and to their honor, the authors discuss the limitations of this study and their differences with other reported studies. Their findings can be useful in terms of clinical manifestations and monitoring of them. 
This Reviewer only has some minor comments, as follow:

1.       Please check throughout the manuscript for spelling mistakes and for sporadically missing abbreviations. In addition, it would be easier for the reader to follow if the use of abbreviations (in specific sentences) could be somehow decreased. 

 →Thank you very much for pointing out this important issue. We checked spells, abbreviations, and Figure2B and changed in red color words for readers to understand results of this study more easily.

2.       Do the authors have more RBC measurements (e.g. RBC count etc.) apart from cellular Hb? It would be very informative in the anemia context of these subjects.

→Thank you very much for pointing out this important issue. We added RBC, MCV measurements in both Table1,and Table2

3.       My main concern is the small number of antibody-positive subjects that can jeopardize the final statistical outcome. In fact, some of the differences are a bit marginal (e.g. p=0.044) nevertheless significant. Can this be attributed the small number of the specific group only or do the authors have another explanation (clinical or statistical)? Please further comment about this in the discussion section.

→Thank you very much for pointing out this important issue. We added and changed to " Finally, the frequency of anti-EPOR antibody-positive patients and those with LVEF<50% was small. As a result, known or possible covariates associated with cardiac dysfunction might not be fully included in the multivariable analysis. In addition, some of the statistical differences are a bit marginal nevertheless significant, and this may be attributed to the small number of the specific group only." in the Discussion limitation (page 8-9, line 281-285).
